# Rotation and Asymmetry of the Axial Plane Pelvis in Cerebral Palsy: A CT-Based Study

**DOI:** 10.3390/children11010063

**Published:** 2024-01-02

**Authors:** Akbar N. Syed, Jenny L. Zheng, Christine Goodbody, Patrick J. Cahill, David A. Spiegel, Keith Baldwin

**Affiliations:** Division of Orthopaedics, The Children’s Hospital of Philadelphia, Philadelphia, PA 19104, USA; syeda@chop.edu (A.N.S.); jenny.zheng@northwestern.edu (J.L.Z.); goodbodyc@chop.edu (C.G.); cahillp1@chop.edu (P.J.C.); baldwink@chop.edu (K.B.)

**Keywords:** cerebral palsy, pelvis, CP pelvis, cobb angle, neuromuscular scoliosis, axial plane deformity

## Abstract

Spinopelvic malignment is commonly seen with non-ambulatory cerebral palsy (CP). Axial plane deformation is not well described in the literature. The purpose of this study was to describe and quantify the axial plane deformity in CP using CT scans and compare it to normal controls. We retrospectively collected data using CT scans of the abdomen and pelvis of 40 patients with GMFCS IV/V CP and neuromuscular scoliosis (CPP) and normal controls (NP) matched by age and sex. Pre-operative Cobb angle was recorded for the CP patients. Pelvic anatomy was evaluated at the supra-acetabular region of bone using two angles—iliac wing angle and sacral ala angle, measured for each hemipelvis. The larger of each hemipelvis angle was considered externally rotated while the smaller angle was considered internally rotated, termed as follows—iliac wing external (IWE) and internal (IWI); sacral ala external (SAE), and internal (SAI). Differences were noted using an independent t-test while correlations with Cobb angle were performed using Pearson’s correlation. Iliac wing measurements showed the externally rotated hemipelvis showed a significantly greater magnitude compared with normal controls at 47.3 ± 18.1 degrees vs. 26.4 ± 3.7 degrees in NP (*p* < 0.001) while no internal rotation was observed (*p* > 0.05). Sacral ala measurements showed greater magnitude in both external and internal rotation. SAE was 119.5 ± 9.5 degrees in CPP vs. 111.2 ± 7.7 degrees in NP (*p* < 0.001) while SAI was 114.1 ± 8.5 degrees in CPP vs. 107.9 ± 7.5 degrees in NP (*p* = 0.001). In the CP cohort, the mean Cobb angle was 61.54 degrees (*n* = 37/40). Cobb angle correlated with the degree of external iliac wing rotation—IWE (r = 0.457, *p* = 0.004) and degree of absolute difference in the rotation of the iliac wing (r = 0.506, *p* = 0.001). The pelvis in a patient with CP scoliosis is asymmetrically oriented exhibiting a greater external rotation of one hemipelvis relative to normal controls. The severity of neuromuscular scoliosis is related to the pelvic axial rotation in CP patients. Axial plane deformity exists in the CP pelvis and this deformity warrants consideration when considering spinopelvic instrumentation strategies and outcomes of supra-pelvic and infra-pelvic pathologies.

## 1. Introduction

Cerebral palsy (CP) is a complex heterogenous condition due to a non-progressive injury to the developing brain. In addition to neurological impairments including abnormalities in muscle tone (spasticity, dystonia), excessive or hyperkinetic movements, and abnormal motor control and weakness, there are also a host of musculoskeletal impairments such as muscle contractures, torsional bone deformities, hip displacement, and scoliosis [1]. While the central nervous system lesion remains static, musculoskeletal deformities such as scoliosis may be progressive [2]. Musculoskeletal deformities may result from an imbalance of muscle forces, abnormalities in muscle tone, positioning, and/or other factors. Patients with greater degrees of neurological involvement, especially those who are non-ambulatory (GMFCS 4-5), will commonly develop a progressive scoliosis. These curves are typically in the thoracolumbar or lumbar region and are associated with pelvic obliquity [2,3,4]. Pelvic obliquity (PO) represents malalignment in the coronal plane, and different techniques have been utilized to quantify this radiographically [5,6]. PO may be associated with supra-pelvic (scoliosis) and/or infra-pelvic (hip displacement, contractures) deformities and may result in considerable impairments in sitting balance/comfort and ease of care [1,5]. Axial plane deformities have received less attention in the literature. We have anecdotally noticed variations in the relationship between the iliac wings and the sacrum when treating CP patients surgically, when using navigation for S2 Alar-Iliac screws, and have sometimes identified an axially “windswept pelvis” [7]. This impacts the technique used when placing these screws or other forms of pelvic fixation.

The axial skeleton has been shown to have structural asymmetry of the hemipelvis in response to mechanical loading during growth and development owing to the bones being more biologically plastic, and an imbalance in mechanical loading can lead to gross asymmetry [8,9,10]. It stands to reason that the abnormal forces seen by spastic muscles in CP would render an axial deformity in the rotation of the hemipelvis in non-ambulant CP. Axial plane deformities of the pelvis may be difficult to appreciate on two dimensional plain radiographs, making the use of advanced imaging like computed tomography (CT) scans essential to better understand the true nature of the deformity [10]. Ko and Sponseller et al. have shown pelvic axial asymmetry in CP patients using CT scans using five measurements with respect to a transverse reference line in the sacrum—(1) upper iliac wing angle (angle between a line extending from the anterior superior iliac spine to the posterior superior iliac spine and a transverse reference line in the sacrum); (2) lower iliac wing above the sciatic notch (angle between the ilium and a transverse reference line in the sacrum); (3) lower iliac wing angle below the sciatic notch (angle between the ilium where the ilium forms an oval on the CT image and a transverse reference line in the sacrum); (4) sacroiliac joint angle (angle measured at the widest level of the sacroiliac joint) and (5) acetabular anteversion (angle between a line connecting the anterior-posterior edges of the widest portion of the acetabulum and the transverse line of the sacrum) [7]. These authors reported that the supracetabular region had the greatest amount of deformity. These authors described a pre-defined pelvic asymmetry as a difference of more than 10 degrees between left and right hemi-pelvis and lacked rotational description [7]. True morphological change may be better understood by quantifying rotation in this population versus a normal control group and assessing any rotation using an axis of rotation. This has been shown in previous studies in adolescent idiopathic scoliosis which have used 3D and 2D CT scans to assess the rotation of the vertebral body and pedicle secondary to the spinal rotation [11,12,13].

Therefore, the purpose of this study was to quantify the axial rotation of the pelvis (CPP) among children and young adults (<25 years) with CP compared to the normal control pelvis (NP). CT scans were used to assess the iliac wing and sacral ala rotation, building upon the work of Ko and Sponseller et al. to further define this phenomenon. We hypothesized that axial plane deformities of the pelvis, manifesting as asymmetry in the relationship between the iliac wings and the sacrum, are common in non-ambulatory patients with cerebral palsy (GMFCS 4-5) and that the deformity correlates with the degree of neuromuscular scoliosis assessed using the Cobb angle.

## 2. Materials and Methods

An institutional review board approved retrospective review was conducted from 1 January 2015 to 31 December 2022 at a single tertiary care pediatric institution. Using CPT codes 74176, 74177, and 74178, we identified CT scans of the abdomen/pelvis of pediatric and young adult patients (<25 years). Patients were included if there was a diagnosis of cerebral palsy and neuromuscular scoliosis with GMFCS Level 4 or 5 (non-ambulatory). Patients with other causes of neuromuscular scoliosis, prior hip or pelvic surgery, history of pelvic/hip fracture, or underlying osseous pathology were strictly excluded. A control group of patients without CP were identified if they had undergone a CT scan without documented radiologic bony pathology (negative study) and were matched by age and sex. The majority of patients in the control group underwent a CT scan for non-orthopedic-related abdominal pain. Control patients were excluded if they had any subjective or objective documented history of a pelvic fracture, deformity, or underlying osseous pathology.

Axial pelvic asymmetry was evaluated with respect to the supra-acetabular region of bone using two landmarks for the iliac wing and sacral ala each. We identified two landmarks for iliac wing which utilized the supraacetabular osseous corridor connecting the anterior inferior iliac spine and the posterior superior iliac spine. For the sacral ala, the highest points of the medial and lateral peak of each hemi-sacrum were used. We used the highest points of the sacral ala as these may represent the rotational change brought about by the imbalance of muscular forces that may produce uneven medial and lateral peaks for each hemi-sacrum. The deformity was measured using two angles—the iliac wing angle and the sacral ala angle. The aforementioned landmarks for the iliac wing and sacral ala were checked for consistency in each axial slice to best represent them as fixed points in each hemipelvis. This was performed to eliminate or minimize the effects of sagittal and coronal plane obliquity using the “copy to all” function on ISITE (Phillips Imaging Inc.). First, we measured the angle between a line bisecting the sacrum and a line drawn from the midportion of the posterior ilium to the midportion of the anterior inferior iliac spine (ASIS) referred to as the iliac wing angle to assess the iliac wing asymmetry (Figure 1). A second angle was measured between a line across the highest points of the sacral ala and the sacral bisector to assess the sacral ala asymmetry, termed the sacral ala angle (Figure 2). The larger of the two angles was considered externally rotated while the smaller angle was considered internally rotated. For each hemipelvis, external/internal rotation (ER/IR) of the iliac wing and sacral ala were termed iliac wing external (IWE)/iliac wing internal (IWI) and sacral ala external (SAE)/sacral ala internal (SAI) respectively. Absolute differences in ER/IR of the iliac wing and sacral ala for one hemipelvis with the corresponding contralateral hemipelvis of the same subject were termed as iliac wing angle difference and sacral ala angle difference, respectively. Additionally, if the patients with cerebral palsy and neuromuscular scoliosis had a full spine AP radiograph available for the measurement of Cobb’s angle (taken prior to any spinal instrumentation), it was recorded. All radiological measurements were reviewed by two authors (A.N.S and J.Z) of this study. To determine inter-observer reliability, two authors reviewed all the angles and were blinded to each other’s measurements. The first author’s measurements were used for analysis.

CPP angle measurements and their difference were compared to matched NP. Descriptive statistics were used to report demographic characteristics for both the cerebral palsy group and the control group. Interrater reliability for the axial pelvic asymmetry variables was assessed using the intraclass correlation coefficient (ICC) based on absolute agreement and classified using the Landis and Koch criteria [14]. After ensuring data normality using the Shapiro–Wilk test, an independent-sample student t-test was used to compare all the measurements between the two angles. Correlation of the Cobb angle with IWE, IWI, SAE, SAI and absolute differences in ER/IR of iliac wing and sacral ala for one hemipelvis was performed using Pearson’s correlation. All tests were conducted using SPSS Version 28 (SPSS Inc., Chicago, IL, USA). Alpha was set at *p* < 0.05 for significance.

## 3. Results

We identified a total of 40 patients to include in each group with a similar mean age of 12.7 years (range, 4–23 years) with only four patients (10%) who were older than 18 years. Furthermore, 95% of the CP cohort were GMFCS level 5 (38/40). In the CPP group, 21 patients had spinal instrumentation at the time the CT-scan was obtained, with 14 patients undergoing posterior spinal fusion and seven undergoing implantation of vertical expandable prosthetic titanium rib. The demographic characteristics for the cerebral palsy group and control group are outlined in Table 1. In the cerebral palsy group, the mean Cobb angle or pre-operative Cobb angle for operative cases was 61.54 degrees (range, 7–119) (*n* = 37/40). In the cases of the patients with spinal instrumentation, spine radiographs used for measuring Cobb angle had a mean time of 4.9 months (SD: 11.6, range 0.03–51.4) between the performance of the radiograph and the CT scan. Inter-rater reliability tests showed strong agreement for both iliac wing and sacral ala angles for single measures with an ICC of 0.87 (95% CI = 0.831–0.906) and 0.86 (95% CI = 0.814–0.869), respectively.

Iliac wing measurements showed greater external rotation in the cerebral palsy group with IWE at 47.3 ± 18.1 degrees (range, 21–87.2) in the CPP group vs. 26.4 ± 3.7 degrees (range, 18.4–36.2) in the NP group (*p* < 0.001). The internally rotated side in the CPP group was similar to controls with IWI at 25.5 ± 11.4 degrees (range, 2.7–52.3) in the CPP group vs. 23.6 ± 4 degrees (range, −14.4–33.9) in the NP group (*p* = 0.170). Iliac wing angle difference of the external and internal hemipelvises in the CPP group was 21.8 ± 20.7 degrees (range, 0.2–69.8) vs. 2.9 ± 2.2 (range, 0–8.4) in the NP group with respect to the hemipelvis of CP patients and controls (*p* < 0.001).

Sacral ala measurements showed differences in both internal and external rotation with SAE at 119.5 ± 9.5 degrees (range, 99.8–140.8) in the CPP vs. 111.2 ± 7.7 degrees (range, 98.1–128.6) in the NP (*p* < 0.001) while SAI was 114.1 ± 8.5 degrees (range, 92.4–126) in the CPP vs. 107.9 ± 7.5 degrees (range, 95.2–123.1) in the NP (*p* = 0.001). Differences seen in the sacral ala angle between the each hemipelvises of individual NP and CPP were 3.2 ± 2.4 degrees (range, 0.2–26.4) vs. 5.4 ± 6.3 degrees (range, 0–10), respectively (*p* = 0.022).

In the CPP group, Cobb angle correlated with the degree of external iliac wing rotation—IWE (r = 0.457, *p* = 0.004) and iliac wing angle difference in the rotation of the iliac wing (r = 0.506, *p* = 0.001). No correlation of Cobb angle was observed in the degree of absolute difference in the rotation of the sacral ala nor with SAE, SAI or IWI (*p* > 0.05).

Two comparative case examples of a 11 year old and 20 year old CP patient with their respective controls has been depicted in Figure 3 to demonstrate the rotation change in the CP pelvis.

## 4. Discussion

Axial plane asymmetry in the CP population has been reported in the literature but is not fully understood [7,15]. Our study builds upon previous work examining the axial plane rotation of the pelvis among 40 non-ambulatory pediatric and young adult patients with CP and compares them with age and sex matched controls. With a majority of patients being GMFCS level 5 patients (95%), our findings suggest the presence of structural external rotation of one hemipelvis in the axial plane compared to the normal controls is associated with the severity of scoliosis with the rotational asymmetry driven more by the iliac wing than the sacral ala. Additionally, the iliac wing showed a large external rotation and no significant contralateral internal rotation compared to normal while the sacrum showed a small external and internal rotation.

The growing pelvis is biologically plastic and exhibits some degree of normal pelvic asymmetry during skeletal growth in response to mechanical loading which is influenced by various factors—gender, handedness, morphogenetics, environment and nutrition [8]. However, in the setting of an abnormal unilateral or asymmetric mechanical force, a significant pathological morphologic change may arise leading to deformities [8,9,16,17]. Such abnormal muscular forces and mechanical loading arise from asymmetrical positioning or posture in early life and has been suggested as a cause for developing neuromuscular scoliosis and hip dislocations which then is compounded over time in children with CP [18,19]. Similarly, with respect to the “non-CP” population, asymmetric mechanical or muscular forces have been shown to produce rotational change in the entire innominate bone or changes in femoral anteversion in the developmental dysplasia of the hip and of the vertebrae or the pedicles in adolescent idiopathic scoliosis [11,12,20,21]. In our study, we assessed the morphological change brought about by asymmetric mechanical forces acting on the pelvis with the relative contributions of the iliac wing and the sacral ala using various measurements to examine asymmetry (iliac wing angle difference and sacral ala angle difference), internal rotation (IWI and SAI) and external rotation (IWE and SAE). In light of the differences seen in the iliac wing and sacral ala measurements while also noting the rotational morphological changes observed in developmental dysplasia of the hip and adolescent idiopathic scoliosis, we postulate that an imbalance of biomechanical forces directed at the CP pelvis may be responsible for this axial bony deformity. Furthermore, this may be compounded over time as seen in neuromuscular scoliosis.

Iliac wing orientation was assessed at the supra-acetabular region termed as the supra-acetabular osseous corridor which is represented by the thickest column of the iliac bone extending from ASIS to the posterior ilium as thicker bone shows greater resistance to torsional change. Hence, any changes observed here should be germane to the remainder of the axial pelvic alignment [22,23]. Iliac wing orientation of the right and left hemipelves in non-ambulatory CP patients was noted to be asymmetric in the present study as demonstrated by the mean absolute difference in the iliac wing angle between the CP patients (21.8 degrees) compared to the controls (2.9 degrees). Ko et al. examined the pelvic asymmetry versus controls at various levels of the right versus left iliac bone and reported a difference in mean angular measurements for iliac orientation in CP patients versus controls and noted differences in the numeric means of the upper iliac angle (12 degree vs. 3 degrees), lower iliac angle above sciatic notch (11.6 degree vs. 3.5 degree), and lower iliac angle below sciatic notch (13.3 degree vs. 3.7 degree) [7]. However, the study reported on the rotation of the individual side, i.e., left hemipelvis versus right hemipelvis and not with respect to the relative orientation of the contralateral side with respect to normal. Muscular imbalances and poor postural control create a unilateral pelvic lift to produce a low-lying ipsilateral hemipelvis and high-lying contralateral hemipelvis [15], which may ultimately lead to an asymmetry and abnormal passive rotational adaptation of the innominate bone under conditions of asymmetric biomechanical loading [16]. With respect to rotation, we noted the iliac wing to show only increased external rotation (26.4 degrees vs. 47.3 degrees) and no difference in internal rotation (23.6 degrees vs. 25.5 degree) when comparing CP patients to normal. The observation of isolated external rotation of the iliac wing may be related to the previously described ipsilateral low-lying pelvis undergoing an internal rotation which is limited by the intra-pelvic contents while producing an un-inhibited external rotation of the contralateral high lying pelvis. In addition, the bony and ligamentous supports of the pelvis may play a role as the posterior structures more strongly resist internal rotation [24]. Current literature has not examined the iliac wing rotation using well-defined measurements. However, Ko et al. reported an inward iliac orientation of one side and pronounced deviation above the acetabulum in severely windswept hips without using angles to quantify the inward rotation [7].

Sacral asymmetry of a small degree has been shown to exist among normal individuals as reported by anatomical studies and is possibly generated from asymmetric loading during growth largely associated with dominant side or handedness [9]. Sacral asymmetry has not been examined in pathological states like CP, and our study is the first to describe unilateral asymmetric mechanically loaded (through sitting on an oblique pelvis) sacrum to describe the pathological sacral supraacetabular morphology which was assessed using highest medial and lateral points of the hemi-sacrum. We noted significant sacral asymmetry (5.4 degrees versus 3.2 degrees), external (119.5 degrees versus 111.2 degrees) and internal rotation (114.1 degrees versus 107.9 degrees) between CP versus controls. Despite the unilateral loading in CP, quantitative deviation of the sacral asymmetry (5.4 degree) was much less than the iliac wing (21.4 degrees) suggestive of its relative resistance to spatial morphological change. This sacral resistance may be attributed to its inherent bony architecture (trabecular patterns) in dissipating biomechanical forces internally and the keystone effect created by its location between the spine and pelvic girdle transmitting nearly all of the biomechanical load to adjacent structures through the sacroiliac joint [25,26,27]. Ko et al. showed numeric differences in the mean sacroiliac angle defined as the angle at the widest level of the sacroiliac joint measured along sacroiliac joint between CP patients (6 degrees) and controls (3.5 degrees) [7]. However, the sacroiliac joint space is relatively dynamic compared to the iliac wing or sacrum which may change with posture and sitting preference and contributes to the discordance between the true asymmetry and measured asymmetry. Future studies should choose reproduceable bony landmarks or dynamic imaging studies to assess the change in sacroiliac joint space. With respect to the degree of asymmetry observed, we hypothesize that the sacrum represents a bony pivot along with the transmitted abnormal mechanical forces that produces some sacral asymmetry while much of the force is transmitted to the iliac wing resulting in larger iliac asymmetry.

Obliquity in the coronal plane (PO) has vastly been the focus of literature surrounding pelvic asymmetry in spastic cerebral palsy which frequently raises the role of muscular imbalances at the spine (supra-pelvic obliquity) or hip (infra-pelvic obliquity) [28,29,30,31]. Lett et al. examined pelvic obliquity, supra-pelvic obliquity and infra-pelvic obliquity suggesting a caudal to cranial or upward sequence of events in a triad sequence wherein dislocation of the hip was followed by pelvic obliquity, and lastly scoliosis [29]. A proportionate relationship between PO with supra-pelvic deformity (scoliosis) and infra-pelvic deformities (hip displacement and asymmetric range of motion in hip abduction) has been observed suggesting an increase in the supra-pelvic deformity or infra-pelvic deformity contributing to an increase in PO [5,7]. Recognizing the lack of literature surrounding the transverse plane asymmetry and its association with supra-pelvic deformity and infra-pelvic deformity, we examined the former, i.e., the degree of neuromuscular scoliosis (supra-pelvic deformity) in relation to the axial plane rotation. We observed a medium magnitude positive correlation of Cobb angle with external rotation of iliac wing (r = 0.457) and absolute differences in the iliac wing rotation (r = 0.506) which suggests the severity of scoliosis may be related to the rotation and asymmetry. However, the rotational effect of the curve complexity, curve type, concomitant infra pelvic factors and deformity in other planes (e.g., pelvic obliquity or sagittal plane deformity) must be further elucidated in future studies. At this point, it is difficult to determine whether the axial pelvic deformity observed is a cause or effect of supra-pelvic deformity and infra-pelvic deformity.

Our retrospective study has several limitations. Subject selection was limited to CT scans available and due to the lack of CT scans in other planes or three-dimension reconstructions, we were only able to present the axial plane rotation. Some CP patients had prior spinal fusion surgery for scoliosis correction at the time of this study which may influence the results presented in this study. However, we do not expect the surgery itself to change the bony morphology of the pelvis in the axial plane over the short term, but continued loading in the setting of residual pelvic obliquity may potentially alter our findings over the long term. Although CT scans provide an accurate and high-resolution depiction of osseous changes, their use is limited due to lack of standardizing supine positioning which poses a challenge in non-ambulatory CP patients due to co-existing supra or infra-pelvic deformities. Lastly, while the results for this study are from a large pediatric institution, their generalizability is limited to non-ambulatory CP patients and may not represent other neuromuscular conditions in which spasticity is present.

Our observations are of value to surgeons treating neuromuscular scoliosis in understanding the deformity for optimal preoperative planning, especially in conditions where intraoperative navigation is not available. Extension of spinal instrumentation to the pelvis is performed in the vast majority of patients with neuromuscular scoliosis, especially those who are non-ambulatory, and have thoracolumbar and lumbar curves associated with significant pelvic obliquity. The most common technique involves placement of S2 Alar-Iliac screws, and the insertion point is typically in between the first and second sacral foramina and just outside their lateral border. The trajectory passes through the sacroiliac joint and into the ilium extending into the supracetabular region. As such, variations in the relationships between the iliac wings and sacrum in the axial plane may alter the screw trajectory and sometimes even require a change in technique. We learned this anecdotally as we routinely use CT-guided navigation to facilitate the placement of these screws. In cases in which the iliac wing is internally rotated, the starting point for the screw must be lateral to the standard location to penetrate the iliac wing, and often requires the screw to be placed through the sacroiliac joint or even start in the posterior ilium bypassing both the sacrum and SI joint. Changes in trajectory are also required when there is external rotation of an iliac wing, and while it may make it easier to start in the sacrum, the ability to engage the ilium may be impacted by the midline structures (sacrum).

The pelvis in the CP patient is asymmetrically oriented exhibiting a greater external rotation of one hemipelvis relative to normal controls. We observed the majority of this deformity in the ilium compared to the sacrum. Additionally, the severity of neuromuscular scoliosis is related to the pelvic axial rotation in CP patients. However, it is also important to note the findings of this study may be influenced by the measurement technique employed as Ko et al. made measurements with respect a transverse line to the sacrum while the present study used a pelvic bisector line to the which measurements were made. While this is a preliminary study built from the foundation of the previous work of Ko and Sponseller, a follow-up study is required to assess the effects of infra-pelvic (hips dislocation, limb contractures) and supra-pelvic deformities (scoliosis) to better assess the axial and appendicular malalignment with their mixed effects. Additionally, it should include an evaluation of contributions from each individual deformity. The management of the non-ambulatory CP patient should consider a thorough evaluation of the pelvis as this may improve sitting imbalance and decrease the surgical burden in the CP patient. Surveillance for hip dislocation and scoliosis detection/progression has been used a preventative measure in CP, however, the pelvic asymmetry is not accounted despite the deformity being described as a triad in CP children. Therefore, we also suggest future studies to develop a clinical algorithm or surveillance strategy to detect or account for the severity of the pelvic deformity. Long-term longitudinal follow-up of individual CP patients is required to assess for variability and rate of progression more accurately. Further understanding regarding the sequence of development of these morphological changes can be achieved through longitudinal study design.

## Figures and Tables

**Figure 1 children-11-00063-f001:**
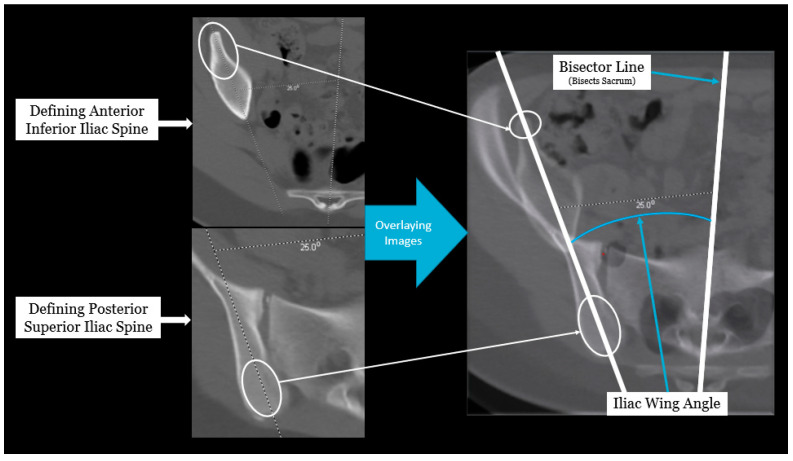
Radiographic measurement technique for iliac wing angle.

**Figure 2 children-11-00063-f002:**
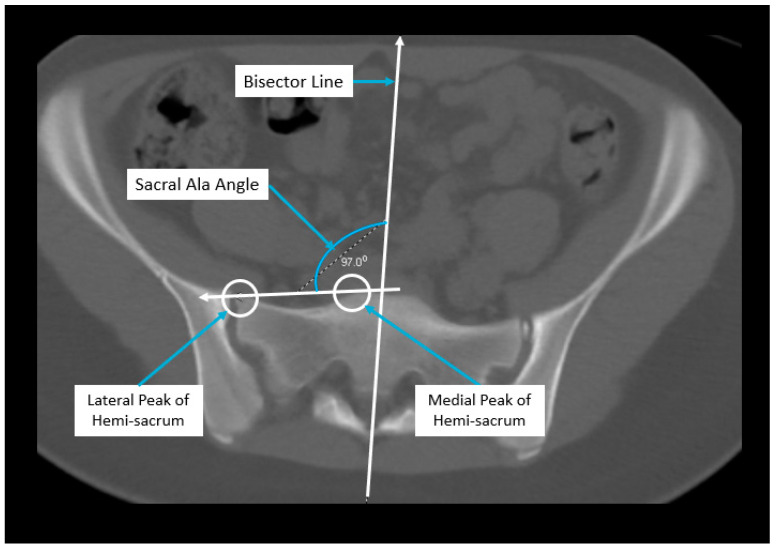
Radiographic measurement technique for sacral ala angle.

**Figure 3 children-11-00063-f003:**
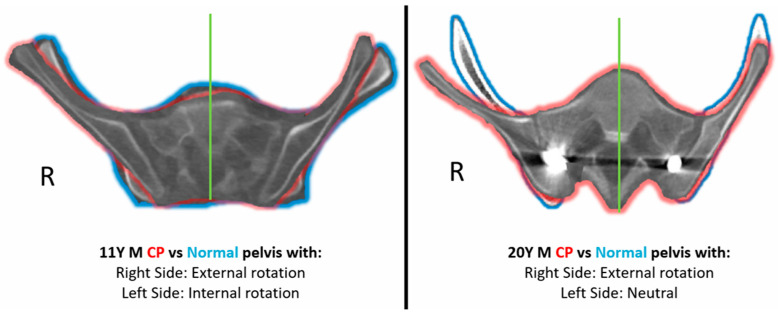
Comparative case example of CP pelvis (Green Outline) with the corresponding matched control pelvis (Blue Outline). Green line represents the sacral bisector.

**Table 1 children-11-00063-t001:** Patient characteristics and measurements.

		Normal Control Pelvis (NP)		Cerebral Palsy Pelvis (CPP)		*p*-Value
		n	Mean (SD)	n	Mean (SD)	
Age (years)		40	12.73 (4.5)	40	12.73 (4.5)	1.000 *
Sex	Male	40	23	40	22	0.822 ^+^
	Female		17		18	
Mean Cobb Angle (Range)		-		37	61.5 (7–119)	-
GMFCS	IV	-		40	2	
	V				38	
Iliac Wing	IWE (degrees)	40	26.4 (3.7)	40	47.3 (18.1)	<0.001
	IWI (degrees)	40	23.6 (4.0)	40	25.5 (11.4)	0.170
Sacral Ala	SAE (degrees)	40	111.2 (7.7)	40	119.5 (9.5)	<0.001
	SAI (degrees)	40	107.9 (7.5)	40	114.1 (8.5)	<0.001
Iliac Wing Angle Difference (degrees)		40	2.9 (2.2)	40	21.8 (20.7)	<0.001
Sacral Ala Angle Difference (degrees)		40	3.2 (2.4)	40	5.4 (6.3)	0.022

+ Chi-square test, * two tailed student’s *t*-test, *p* < 0.05 was considered significant.

## Data Availability

Data available on request due to restriction (e.g., privacy or ethical concerns). The data presented in this study are available on request from the corresponding author.

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
