# Peer review of "Rotation and Asymmetry of the Axial Plane Pelvis in Cerebral Palsy: A CT-Based Study"

_children, 2024, doi:10.3390/children11010063_

Round 1

Reviewer 1 Report

Comments and Suggestions for Authors

The authors provide the first detailed study of both sacral and pelvic rotational deformities in youth with CP compared to age-matched controls. This is a retrospective study with a robust design, the manuscript is well written, and the limitations appropriately acknowledged. I believe it will make a meaningful addition to the literature on scoliosis and pelvic imbalance in youth with CP. Following some minor issues that should be addressed :

P2 li 84 - 86 "axial rotation of the CP pelvis 84 (CPP) among children and young adults (<25 years) compared to the normal control pelvis 85 (NP)". Please use person-centred writing, if I'm OK to keep the CPP acronym the sentence should read something like "... of the pelvis (CPP) of children and young adults with CP...".

Table 1 : I find it very cramped, would deserve a slightly more "aerated" style

P5 li 176 - 178 ipsilateral and controlateral to what? (the spinal curvature?) If not favor "between each hemipelvis"

P5 li 181 - 182 "degree of absolute difference in the rotation of the iliac 181 wing", favor terminology used in methods for coherence i.e. " Iliac Wing Angle Difference"

Reviewer 2 Report

Comments and Suggestions for Authors

1, It is an interesting question, how measure the pelvic obliquity, but the paper should be more clear, and should include the latest references, especially regarding assesing pelvic aligmnment using classic x-ray;

2. design of material/ methods has big weakness: choice of control group - healthy without CP and without scoliosis! Control group should consist of CP patients or scoliosis patients.

3. Authors describe scoliosis - but in which direction (right or left) - in relation to pelvic obliquity?

4. Why there is nothing about hips? May be obliquity is connected both with hips and scoliosis.

5. I do knot konw is the main goal of the paper is description of the new method or results of pelvic obliquity in CP patients?

6. What is new regarding the work of Ko et al?

  •  

Author Response

Reviewer 2:

  • It is an interesting question, how measure the pelvic obliquity, but the paper should be more clear, and should include the latest references, especially regarding assessing pelvic alignment using classic x-ray;
    • We appreciate the reviewer’s insightful comment, however, there are various methods to measure pelvic obliquity and we have sighted the most appropriate journal articles which highlight and assesses different methods of pelvic obliquity.
  • design of material/ methods has big weakness: choice of control group - healthy without CP and without scoliosis! Control group should consist of CP patients or scoliosis patients.
    • We appreciate the reviewer’s thoughtful comment and have considered this. However, being an anatomical study, we chose the controls as the normal human pelvis. With a study aim to describe the axial deformation in CP scoliosis, comparison of CP with CP scoliosis has less meaning as both represent pathological states. By design we would need a cohort without any pathological influence to describe the changes from normal/non-pathologic states.
  • Authors describe scoliosis - but in which direction (right or left) - in relation to pelvic obliquity?
    • We appreciate the reviewer’s comment. While we agree with the reviewer’s comment that the addition of the left and right direction of the scoliosis is appropriate, the current study is only focus on describing the axial deformation phenomenon (not pelvic obliquity) and the effect of scoliosis can be attempted in subsequent paper as we have shown some association with the degree of scoliosis.
  • Why there is nothing about hips? May be obliquity is connected both with hips and scoliosis.
    • We appreciate the reviewer’s excellent suggestion and these are future directions for the paper. However, we are statistically limited by the number of pelvic x-rays in the current cohort to accurately describe this.
  • I do not know is the main goal of the paper is description of the new method or results of pelvic obliquity in CP patients?
  • What is new regarding the work of Ko et al?
    • Reply for point 5 and 6: We thank the reviewer for their comment. The current study is not designed to study the classical pelvic obliquity and we have used the “pelvic obliquity” anecdotally as it a conormal plane deformity. We have designed this paper to describe the axial deformation (not coronal) of the pelvis and build on the work of Ko et al’s literature. They use of a horizontal line across the pelvis to describe the axial deformity (as described in lines 65 – 74) and have not described the sacrum. We have hemisected the pelvis to better describe the phenomenon and also comment on the sacrum. Lastly, Ko et al only compared the left hemi-pelvis to the right hemi-pelvis, however, we describe the rotational profile of the pelvis.

Reviewer 3 Report

Comments and Suggestions for Authors

Very interesting study with good number of patients and follow up. The paper  is well prepared. I would like to pay attention for some points.

The introduction is too long and hard to read it, please shorten, only the most important points.

The result could be better understanding of outcomes with more figures and example of clinical pictures patient underwent surgical treatment.

The discussion should be improved with some newest references about surgical treatment of patients with CP, there have been released several good studies lately.

Author Response

Reviewer 3:

  • Very interesting study with good number of patients and follow up. The paper is well prepared. I would like to pay attention for some points.
  • The introduction is too long and hard to read it, please shorten, only the most important points.
    • We have made the changes per the reviewer. We have omitted a few sentences from the first paragraph in the introduction. However, we have adhered to the minimum word count of 4000 words.
  • The result could be better understanding of outcomes with more figures and example of clinical pictures patient underwent surgical treatment.
    • The reviewer has made an excellent suggestion, however, being an anatomical study, we have included 2 case examples using the CT scans. We also have limited access to clinical pictures.
  • The discussion should be improved with some newest references about surgical treatment of patients with CP, there have been released several good studies lately.
    • We appreciate the reviewer’s comment. The current study is an anatomical one focusing on describing the anatomical variation of the pelvis rather surgical technique. We have used the discussion section to elaborate on the possible causations of this phenomenon. Addition of various surgical technique and their influences maybe attempted in a separate paper that examines effect of scoliosis on this phenomenon. Lastly, addition of surgical treatment may mislead the reader from the study goals.

Round 2

Reviewer 2 Report

Comments and Suggestions for Authors

Thank you for your answers to my comments. 

Regarding new version:

To sum up, introduction - background is fully described, goals of the study clearly stated. Mat.&methods - clearly explained as well as results.

Discussion:

1. lines 330-333: you mention previous work of Ko and Sponseller - it would be great if you underline differences in measurements between them and current paper.

2. it should be more clearly stated that future research of the correlation between pelvic rotational asymmetry, pelvic obliquity, scoliosis and hips would be advisable.
